# The Gut Microbiome Responds Progressively to Fat and/or Sugar-Rich Diets and Is Differentially Modified by Dietary Fat and Sugar

**DOI:** 10.3390/nu15092097

**Published:** 2023-04-27

**Authors:** João Pessoa, Getachew D. Belew, Cristina Barroso, Conceição Egas, John G. Jones

**Affiliations:** 1CNC—Center for Neuroscience and Cell Biology, University of Coimbra, 3004-504 Coimbra, Portugal; joao.pessoa@cnc.uc.pt (J.P.); gechap4@gmail.com (G.D.B.); cristina.barroso@biocant.pt (C.B.); john.griffith.jones@gmail.com (J.G.J.); 2CIBB—Center for Innovative Biomedicine and Biotechnology (CIBB), University of Coimbra, 3004-531 Coimbra, Portugal; 3Biocant-Technology Transfer Association, Biocant Park, 3060-197 Cantanhede, Portugal

**Keywords:** mouse model, diet, high-fat, high-sugar, gut microbiome, DNA sequencing, phylum, genus

## Abstract

Describing diet-related effects on the gut microbiome is essential for understanding its interactions with fat and/or sugar-rich diets to promote obesity-related metabolic diseases. Here, we sequenced the V3-V4 hypervariable region of the bacterial 16S rRNA gene to study the composition and dynamics of the gut microbiome of adult mice fed diets rich in fat and/or sugar, at 9 and 18 weeks of diet. Under high-fat, high-sugar diet, the abundances of *Tuzzerella* and *Anaerovorax* were transiently increased at 9 weeks, while *Lactobacillus* remained elevated at 9 and 18 weeks. The same diet decreased the abundances of *Akkermansia*, *Paludicola*, *Eisenbergiella*, and *Butyricicoccus* at 9 and 18 weeks, while *Intestinimonas* and UCG-009 of the *Butyricicoccaceae* family responded only at 18 weeks. The high-fat diet decreased the abundances of UBA1819 at 9 weeks, and *Gastranaerophilales*, *Clostridia* UCG-014, and ASF356 at 9 and 18 weeks. Those of *Marvinbryantia*, *Harryflintia*, *Alistipes*, *Blautia*, *Lachnospiraceae* A2, Eubacterium coprostanoligenes group, and Eubacterium brachy group were lowered only at 18 weeks. Interestingly, these genera were not sensitive to the high-sugar diet. The mouse gut microbiome was differentially affected by diets rich in fat or fat and sugar. The differences observed at 9 and 18 weeks indicate a progressive microbiome response.

## 1. Introduction

The human gut is populated with a large and diverse bacterial microbiome, whose numbers of cells roughly equal that of the host [1]. It contains about 22 million non-redundant genes, half of them individual-specific [2].

The gut microbiome is a critical intermediary between the diet and its impact on the metabolism of the host. Our group has shown that diets containing glucose or fructose as the only carbohydrate source produced different gut microbiomes in mice [3]. Metabolites produced by the gut microbiome can reach the liver through the portal vein; therefore, any hazardous substances produced by the gut microbiome can exert deleterious effects on this major metabolic organ. Consistently, extensive alterations in the gut microbiome have been detected in patients with non-alcoholic fatty liver disease (NAFLD) [4] and cirrhosis [5]. These conditions are partially caused by alterations in the composition and levels of gut microbiome metabolites, which include short-chain fatty acids such as acetate, propionate, and butyrate [6]. In the gut, propionate and butyrate activate gluconeogenesis through distinct gene expression mechanisms [7]. Furthermore, gut microbiome metabolites cross-talk with dietary products. For example, the conjunction of dietary fructose and gut microbiome-produced acetate enhances hepatic lipogenesis and consequent fat accumulation in the liver [8]. In a study of mice fed a diet whose carbohydrate component was either 100% glucose or 100% fructose, those fed with fructose showed increased deconjugation of bile acids and depleted butyrate and taurine factors that might promote dysbiosis and impair intestinal barrier function [3]. Since the glucose and fructose diets also resulted in different gut microbiomes, these findings indicate that the effects of the fructose diet on the host were in part mediated by its effects on the gut microbiome.

The gut microbiome and its metabolites can also affect other organs, including the liver, heart, lungs, and kidneys [9]. Gut microbiome modulation has been proposed as a potential therapeutic approach against NAFLD [10,11], type 2 diabetes [12,13] and other diseases. These findings indicate that interventions in gut microbiome components hold great therapeutic promise. Recent developments in DNA sequencing technologies have increased the available number of highly accurate bacterial genomic sequences. These advances provide a unique window of opportunity for uncovering the dynamic composition of the gut microbiome in unprecedented detail.

Here, we studied the composition and dynamics of the gut microbiome of adult mice fed with diets enriched in fat and/or sugar. We describe alterations and identify bacterial phyla and genera differentially affected by fat and/or sugar, at 9 and 18 weeks of diet.

## 2. Materials and Methods

### 2.1. Animal Treatments

Animal studies were approved by the University of Coimbra Ethics Committee on Animal Studies and the Portuguese National Authority for Animal Health (Direcção-Geral de Alimentação e Veterinária, DGAV), approval code 0421/000/000/2013. Adult male C57BL/6 mice were obtained from Charles River Labs (Barcelona, Spain) and housed at the UC-Biotech Bioterium (Cantanhede, Portugal). Mice were maintained in a well-ventilated environment, under a 12 h light/12 h dark cycle. Upon delivery to the Bioterium, mice were provided a 2 week interval for acclimation, with free access to water and a standard chow diet comprising 73% mixed carbohydrates, 19% proteins, 4% lipids, and 4% ash. After this period, the mice were divided into four groups. The control group (CTL; *n* = 12) was kept under a standard chow diet, as before. In the high-fat group (HF; *n* = 11), the standard chow diet was replaced by 41% mixed carbohydrates, 25% proteins, 30% lipids, and 4% ash (% by weight). The detailed composition of the CTL and HF diets is listed in Appendix A. In the high-sugar group (HS; *n* = 12), drinking water was supplemented with a 30% (*w*/*v*) mixture of 55/45% fructose/glucose. The high-fat, high-sugar group (HFHS; *n* = 11) corresponded to the conjunction of the high-fat and high-sugar diets. Due to their disruptive behavior, one mouse had to be removed from both the HF and the HFHS diet groups, resulting in slightly different animal numbers among the four experimental conditions. Diets were kept for 18 weeks. Feces were collected at 0, 9, and 18 weeks. Genomic DNA was isolated using the QIAamp DNA Stool Mini Kit (Qiagen, Hilden, Germany), according to the kit protocol.

### 2.2. Sample Preparation and DNA Sequencing

Samples were prepared for Illumina Sequencing by 16S ribosomal RNA (rRNA) gene amplification of the bacterial community. The DNA was amplified for the hypervariable V3-V4 region with specific primers and further reamplified in a limited-cycle PCR reaction to add sequencing adapters and dual indexes. First PCR reactions were performed for each sample using KAPA HiFi HotStart PCR Kit according to manufacturer suggestions, 0.3 μM of each PCR primer: forward primer Bakt_341F 5′–CCTACGGGNGGCWGCAG-3′ and reverse primer Bakt_805R 5′–GACTACHVGGGTATCTAATCC-3′ [14,15] and 12.5 ng of template DNA in a total volume of 25 μL. The PCR conditions involved a 3 min denaturation at 95 °C, followed by 25 cycles of 98 °C for 20 s, 55 °C for 30 s and 72 °C for 30 s, and a final extension at 72 °C for 5 min. Second PCR reactions added indexes and sequencing adapters to both ends of the amplified target region according to the manufacturer’s recommendations [16]. Negative PCR controls were included for all amplification procedures. PCR products were then one-step purified and normalized using SequalPrep 96-well plate kit (Thermo Fisher Scientific, Waltham, MA, USA) [17], pooled and pair-end sequenced in the Illumina MiSeq^®^ sequencer with the V3 chemistry, according to the manufacturer’s instructions (Illumina, San Diego, CA, USA) at Genoinseq (Cantanhede, Portugal).

### 2.3. Sequence Analysis, Plotting, and Statistics

Sequence data were processed at Genoinseq (Cantanhede, Portugal). Raw reads were extracted from an Illumina MiSeq^®^ System in fastq format. The QIIME2 package version 2020.2.0 [18] was used for amplicon sequence variant (ASV) creation with DADA2 [19]. ASV taxonomic assignments were determined with the q2 feature-classifier plugin [20] against the SILVA database version 138 (https://www.arb-silva.de/; accessed on 17 February 2022) [21]. ASVs and corresponding abundance per sample were organized in an abundance table. ASVs not assigned to the *Bacteria* kingdom or assigned to mitochondria or chloroplasts were removed from the abundance table. The abundance table was then used for composition, alpha and beta diversity analysis, and to identify differentially abundant ASVs. Demultiplexed raw sequences are available from the Short Read Archive (SRA) under the accession number PRJNA940685.

Analyses were performed using R Statistical Software (version 4.1.2) [22] (URL: https://www.R-project.org/; accessed on 22 February 2022) in Rstudio version 2021.09.2 build 382 [23] (URL: http://www.rstudio.com/; accessed on 22 February 2022). Alpha diversity measures of richness (Observed, Chao1, and ACE) and diversity (Shannon, Simpson, Inverted Simpson, and Fisher) were calculated with phyloseq version 1.38.0 [24]. Index values were compared using analysis of variance (ANOVA) followed by Dunn’s test or the Kruskal–Wallis test followed by the Mann–Whitney post hoc test, after testing for normality with the Shapiro–Wilk test. Beta diversity was analyzed with principal coordinates analysis (PCoA) using the Bray–Curtis dissimilarity in phyloseq. The difference in microbiome composition between diets and time points was statistically tested with PERMANOVA, followed by pairwise PERMANOVA using the adonis function of the vegan package version 2.5.7 [25] (R package version 2.5-7; URL: https://CRAN.R-project.org/package = vegan; accessed on 22 February 2022). Statistical significance was calculated after adjusting with the Benjamini–Hochberg procedure. PCoA plots were produced with ggplot2 version 3.3.5 [26].

The ASV abundance table was converted into a pivot table using Microsoft Excel 2016. Abundance data of all phyla or genera, under a specific diet and time point, were selected from the pivot table. The percentage of each phylum or genus was calculated by dividing its abundance by the sum of all abundances (including the unclassified reads). The percentages of selected phyla or genera under a specific diet and time point were plotted as violin graphs using GraphPad Prism 8.0.2 (GraphPad Software, San Diego, CA, USA). In each violin graph, dashed and dotted lines represent the median and interquartile ranges, respectively. Differential abundance of taxa between groups was determined with the Analysis of Compositions of Microbiomes with Bias Correction (ANCOM-BC) algorithm [27] version 1.4.0 at phylum and genus levels. *p*-values were calculated after adjusting with the Holm–Bonferroni method.

In all data analyses, an adjusted *p*-value equal to or less than 0.05 was considered statistically significant.

## 3. Results

### 3.1. The Gut Microbiome Changed with Time and Diet, and Its Richness and Diversity Decreased over Time

In this study, we sought to uncover gut microbiome alterations both as a function of diet and time. We utilized four experimental groups of mice. In the control group (CTL), a standard chow diet was maintained throughout the study. In the high-fat diet group (HF), standard chow diet was supplemented with 30% lipids (*w*/*w*). In the high-sugar diet group (HS), drinking water was replaced with a 30% (*w*/*v*) sugar solution. In the high-fat, high-sugar diet group (HFHS), mice were fed the standard chow diet supplemented with lipids to a final concentration of 30% (*w*/*w*), and the drinking water contained 30% (*w*/*v*) sugar. For gut microbiome profiling, fecal DNA was obtained immediately before the start of the dietary alterations (0 weeks), as well as after 9 and 18 weeks of diet (Figure 1A).

For a preliminary assessment of the data, we performed a principal coordinates analysis (PCoA). This analytical tool provides a depiction of global variations within a dataset. PCoA revealed that the microbiome had varied over time, especially between 0 and 9 weeks, regardless of the diet (PERMANOVA F = 5.08, R^2^ = 0.036, *p* = 0.001) (Figure 1B). PCoA also showed that diet had a less significant effect on microbiome composition than time (PERMANOVA F = 4.51, R^2^ = 0.032, *p* = 0.001) (Figure 1C). This preliminary analysis revealed that time itself had a significant effect on gut microbiome alterations. We hypothesize that this time effect was a consequence of mouse aging during the study.

To analyze the diet effects in greater detail, we tracked microbiome variations within each mouse. Regarding the CTL diet, there were roughly two subpopulations: one in which only the first coordinate varied, and another in which both the first and second coordinates were variable (Figure 1D, top). Concerning the HF diet, there were also two subpopulations. The first coordinate varied similarly, while the second coordinate showed two distinct variation types (Figure 1D, middle-top). In the HFHS diet, the two subpopulations were less separated (Figure 1D, middle-down). In the HS diet, the distribution was the most homogeneous, and no subpopulations were identified (Figure 1D, down). These observations suggest that the HS diet induced a more uniform response in the gut microbiome of different mice. Both the HF and CTL diets seemed to induce more heterogeneous responses. The HFHS diet could have induced an intermediate response between the HF and HS diets, with respect to overall gut microbiome alterations.

To search for additional effects in the gut microbiome, we also assessed microbiome richness, through the calculation of the Observed, Chao1, and ACE indices, for each diet and time point. All microbiome richness indices showed similar trends, and we selected Chao1 as a representative one. It indicated that microbiome richness was fairly homogenous at 0 weeks, but decreased at 9 and 18 weeks of diet (Figure 1E). We also assessed microbiome diversity through the calculation of the Shannon, Simpson, Inverted Simpson, and Fisher indices. The Fisher index showed the largest number of statistically significant differences in microbiome diversity. It indicated that, similarly to microbiome richness, its diversity also followed a similar decrease trend (Figure 1F). The changes between 0 and 9 weeks were statistically significant for both microbiome richness (Appendix A) and diversity (Appendix A). These observations show that time itself induced a decrease in both the overall levels and heterogeneity of the gut microbiome. These effects were most obvious between 0 and 9 weeks, suggesting stabilization of the microbiome between 9 and 18 weeks.

### 3.2. Time Had a Major Impact on the Gut Microbiome, but Diet Also Impacted Its Evolution

To further explore microbiome alterations as a function of time and diet, we used pairwise PERMANOVA to identify specific pairs of statistically significantly different groups. Considering time as a factor, the gut microbiome showed statistically significant changes from 0 to 9 weeks within all diets. Nevertheless, from 9 weeks to 18 weeks, we could only observe one statistically significant difference, which was related to the HS diet (Appendix A). Considering diet as a factor, only the microbiomes of HF and HFHS diet mice were not significantly different from each other (Appendix A). As time had a significant effect on the gut microbiome in the CTL group, as well as on diet groups, we tested if the combined effects of time and diet were driving the alterations in the gut microbiome. No differences were observed between all diet pairs at 0 weeks (*p* > 0.05; Appendix A), indicating that initial gut microbiomes were similar between the groups, as expected. We then tested if diet conditioned the gut microbiome alterations. At 9 weeks, all pairwise comparisons indicated different gut microbiomes between diets (*p* < 0.05), except for HFHS, which did not differ significantly from HF (*p* > 0.05; Appendix A). At 18 weeks, the gut microbiomes of all diet groups differed from each other (*p* < 0.05; Appendix A). The observed differences indicated that, although time had a relevant impact on the gut microbiomes, diet also impacted gut microbiomes over time.

### 3.3. Most Bacterial Phyla and Several Genera Remained Stable under the Control Diet

The major goal of the present study was to describe the time variations of bacterial phyla and genera that were altered by different diets. As such, we needed first to identify which phyla and genera were unchanged over time under the CTL diet.

We determined *p*-values corresponding to alterations of bacterial phyla between 0 and 9 weeks, and between 9 and 18 weeks, under the CTL diet (Appendix A). While the percentages of Desulfobacterota were not constant between 0 and 18 weeks, Cyanobacteria and Proteobacteria did vary under the CTL diet only between 0 and 9 weeks.

Then, we repeated the statistical analysis for the same time comparisons at the genus level (Appendix A). From here, we also selected the genera without significant variations over time, for assessing alterations under different diets. Genera with statistically significant variation over time in the CTL diet were also considered; however, this aspect will be mentioned in the subsequent sections.

### 3.4. At 9 Weeks, the HF Diet Changed the Percentages of Several Phyla, and HS Diet Decreased the Percentage of Verrucomicrobiota

After assessing global variations, we looked for more specific alterations. First, we analyzed variations in phyla at 9 weeks of diet. The most abundant phyla, Actinobacteriota, Firmicutes and Verrucomicrobiota, showed distinct alterations. Actinobacteriota increased its percentage under the HF and HFHS diets, with no effect of HS diet alone (Appendix A). Firmicutes, which had a slightly skewed distribution under the control diet, became more homogeneous under the HF diet and showed a different skewing under the HFHS diet (Appendix A). The percentages of Verrucomicrobiota were drastically decreased under the HF and HS diets and their combination (Appendix A). Regarding less abundant phyla, Cyanobacteria were drastically decreased under the HF and HFHS diets, with no effect of HS alone (Appendix A). Proteobacteria showed similar behavior, although the HF diet alone resulted in no statistically significant difference (Appendix A). Nevertheless, the variations of Cyanobacteria and Proteobacteria should be considered carefully, as their control conditions were not unchanged between 0 and 9 weeks (Appendix A).

These observations show that several phyla were affected by the HF diet, with diverse effects, including increase (Actinobacteriota), redistribution (Firmicutes), and decrease (Verrucomicrobiota and Cyanobacteria). These observations suggest that HF levels in the gut can interfere with distinct cellular processes in distinct bacterial phyla. On the other hand, the HS diet only had one clear effect, in only one phylum. It decreased the percentage of Verrucomicrobiota, a phylum with a few species.

### 3.5. At 9 Weeks, the HFHS Diet Decreased the Percentages of Akkermansia, Paludicola, Eisenbergiella, and Butyricicoccus

Then, we analyzed variations in genera at 9 weeks of diet. We identified genera whose percentages decreased under the HF diet. Of the Verrucomicrobiota phylum, the *Akkermansia* genus was essentially eliminated from the gut microbiome under the HFHS diet. The HF and HS diets had drastic effects on this genus, although the HF-related decrease was not statistically significant (Figure 2A). Other genera with similar responses were the lowly abundant *Paludicola* (Figure 2B), *Eisenbergiella* (Appendix A), and *Butyricicoccus* (Appendix A). Of note, the CTL diet showed variations between 0 and 9 weeks for both *Eisenbergiella* and *Butyricicoccus* (Appendix A); nevertheless, their elimination from the gut microbiome under the HFHS diet appeared to be relevant. With the exception of *Akkermansia*, these genera belong to the Firmicutes phylum and their decrease induced by the HFHS diet was sustained at 18 weeks of diet.

### 3.6. At 9 Weeks, the HF (but Not HS) Diet Decreased the Percentages of Gastranaerophilales, Clostridia UCG-014, and UBA1819

Regarding other genera, the HF (but not the HS) diet could induce a decrease in abundance in the gut. The percentage of the *Gastranaerophilales* genus (Cyanobacteria phylum) was significantly reduced by the HF and HFHS diets, but not by the HS diet (Appendix A). Sequencing data also revealed bacterial sequences that did not correspond to well-studied genera. Both *Clostridia* UCG-014 (Appendix A) and UBA1819 (Figure 2C) (both from Firmicutes) were decreased by the HF, but not by the HS diets. Despite being also decreased under the HFHS diet, the differences were not statistically different from the respective controls. These findings suggest that HF interferes more critically with the metabolism of these bacterial genera than HS. With the exception of UBA1819, the HF effects were sustained at 18 weeks of diet. Of note, both *Gastranaerophilales* and *Clostridia* UCG-014 showed time variations under the CTL diet between 0 and 9 weeks (Appendix A).

### 3.7. At 9 Weeks, the HFHS Diet Increased the Percentages of Lactobacillus, Tuzzerella, and Anaerovorax

On the other hand, the percentages of other lowly abundant Firmicutes genera were increased by the combined effect of the high-fat and high-sugar diets. *Lactobacillus* (Figure 2D) and *Tuzzerella* (Figure 2E) were statistically significantly increased by the combined effect of the HFHS diet. Nevertheless, the percentage of the *Anaerovorax* genus was already increased by the HF and HS diets separately (Figure 2F). These findings indicate that the HS diet can have a beneficial effect on the metabolism of these bacterial genera. With the exception of *Lactobacillus*, these alterations were not detected at 18 weeks, indicating that their effect was transient.

### 3.8. At 9 Weeks, the HS (but Not HFHS) Diet Increased the Percentages of ASF356 and Clostridium Sensu Stricto 1

Two other Firmicutes genera were increased by the HS diet, with no detectable effects of HF. In these two genera, the lowly abundant ASF356 (Figure 2G) and the highly abundant *Clostridium* sensu stricto 1 (Figure 2H), this effect was not observed under HFHS, suggesting the HF diet is able to inhibit the effects of HS. In *Clostridium* sensu stricto 1, the increase induced by the HS diet was sustained at 18 weeks.

### 3.9. At 18 Weeks, the HS Diet Increased the Percentage of Previously Unaffected Patescibacteria

The diets were continued for another 9 weeks, and microbiome changes were monitored as before. Of the major phyla, Actinobacteriota and Verrucomicrobiota still showed prominent alterations. The percentage of Actinobacteriota was higher under the HF diet, with no clear changes under the HS and HFHS diets (Appendix A). The percentage of Verrucomicrobiota decreased under the HF, HS, and HFHS diets, although the HF-related decrease was not statistically significant (Appendix A). Concerning the percentages of less abundant phyla, in Cyanobacteria, the trend identified at 9 weeks was maintained. HF and HFHS (but not HS) could lower the percentage of this phylum in the gut microbiome (Appendix A). Proteobacteria was also decreased under the HFHS diet, with a non-statistically significant decrease observed under HF (Appendix A). The percentage of Patescibacteria had a non-statistically significant decrease under the HF and HS diets, and a statistically significant increase under the HS diet (Appendix A), which was not observed at 9 weeks of diet.

In comparison with the alterations observed at 9 weeks, the alterations at 18 weeks indicated similar effects of the HF diet; however, HS-related effects became more prominent. Specifically, in Actinobacteriota, the combined effect of HFHS was no longer statistically different from the control (Appendix A), as it was during the first 9 weeks of diet (Appendix A). This was likely due to the effect of HS, which, although not statistically significant by itself, could counteract the effect of HF during the second period of 9 weeks of diet. Additionally, the Patescibacteria phylum showed an increase in its percentage under the HS diet (Appendix A), which was not statistically significant before. These observations indicated a slower effect of HS in microbiome alterations, at the phylum level.

### 3.10. At 18 Weeks, the HFHS Diet Decreased the Percentages of Previously Unaffected Intestinimonas and UCG-009 of the Butyricicoccaceae Family

We then analyzed again the alterations in the percentage of genera at 18 weeks of diet. The percentage of the highly abundant *Akkermansia* phylum was still reduced under the three diets, although under the HF diet, the decrease was not statistically significant (Figure 3A). An identical behavior was observed in the less abundant *Paludicola* (Figure 3B), for which a statistically significant decrease under the HS diet was observed only at 18 weeks, but not at 9 weeks of diet. *Eisenbergiella* showed statistically significant decreases under the three diets (Figure 3C), despite only having a statistically significant difference for HFHS at 9 weeks (Appendix A). The alterations of these two genera showed that the combined effect of the HFHS diet had a quicker, but overall similar, effect than the HF or HS diets alone. Only the combined effect of the HFHS diet could decrease the percentage of *Butyricicoccus* (Figure 3D), which maintained the trend observed at 9 weeks (Appendix A). *Intestinimonas* showed an identical profile at 18 weeks (Figure 3E). UCG-009 of the *Butyricicoccaceae* family was also decreased under the HFHS diet (Figure 3F). While the alterations in *Akkermansia*, *Eisenbergiella*, *Paludicola*, and *Butyricicoccus* were already observed in the first 9 weeks (Figure 2A–D), alterations in the other genera were only significant at 18 weeks. These findings indicate that bacterial adaptation is progressive, and new events can be detected in more advanced stages.

### 3.11. At 18 Weeks, the HF (but Not HS) Diet Decreased the Percentages of Previously Unaffected Marvinbryantia, Harryflintia, Alistipes, Blautia, Lachnospiraceae A2, Eubacterium Coprostanoligenes Group, and Eubacterium Brachy Group

Four other genera were statistically significantly decreased by the HF diet alone. *Gastranaerophilales*, which were previously observed to be decreased, remained low (Figure 4A). At 18 weeks of diet, the percentages of *Marvinbryantia* (Figure 4B), *Harryflintia* (Figure 4C), and *Alistipes* (Figure 4D) were statistically significantly decreased by the HF and HFHS diets. The percentage of *Blautia* in the microbiome was lowered by the HF diet, being the effect rescued by the HS diet (Figure 4E), an effect not observed at 9 weeks. Concerning uncultured bacterial genera, significant alterations were found only at 18 weeks of diet. At 9 weeks, *Clostridia* UCG-014 was decreased only under the HF diet (Appendix A); however, at 18 weeks, it was also decreased by the combination of HFHS (Figure 4F). UBA1819 lost its statistically significant decrease under the HF diet at 18 weeks. ASF356, which was decreased by the HF and HFHS diets and increased by the HS diet at 9 weeks (Figure 2G), lost the latter change at 18 weeks (Figure 4G). The Eubacterium coprostanoligenes group (Figure 4H) and Eubacterium brachy group (Figure 4I), previously unaffected, were also decreased by the HF and HFHS diets at 18 weeks. *Lachnospiraceae* A2, also not affected at 9 weeks, showed a similar trend, but the HFHS-related alteration was not statistically significant (Figure 4J). These observations indicate that some genera respond more slowly to an HF-rich environment than others. Importantly, these genera (Figure 4A–J) were insensitive to the HS diet. These findings indicate that the HF diet could induce significant alterations in more bacterial genera than the HS diet.

### 3.12. At 18 Weeks, the HF Diet Increased the Percentage of Lactobacillus (Previously Affected by HFHS), and the HFHS Diet Increased the Percentage of Clostridium Sensu Stricto 1 (Previously Affected by HS Only)

At 18 weeks of diet, the percentage of *Lactobacillus* remained elevated under the HF diet and especially under the HFHS diet, without a clear effect of the HS diet (Figure 5A). This behavior was already observed at 9 weeks (Figure 2D), although the HF diet alone could increase this genus only at 18 weeks. On the other hand, *Clostridium* sensu stricto 1 was increased by the HS and HFHS diets (Figure 5B), despite being increased only by the HS diet at 9 weeks (Figure 2H). This alteration suggests that the potential inhibitory effect of the HF- over the HS-related increase in *Clostridium* sensu stricto 1 was transient, lasting for less than 18 weeks.

### 3.13. At 18 Weeks, the HS Diet Decreased the Percentages of Previously Unaffected Ruminococcaceae Incertae Sedis and Lachnospiraceae NK4A136 Group (the Latter Was Simultaneously Increased by HF)

At 18 weeks of diet, additional genera were decreased by the HS diet, including *Ruminococcaceae* Incertae sedis (Figure 5C). The HS diet also decreased the percentage of the *Lachnospiraceae* NK4A136 group; however, the HF diet could also increase this genus. Under the HFHS diet, there was a cancellation effect (Appendix A). It should also be taken into consideration that the *Lachnospiraceae* NK4A136 group was not fully stable under the control diet between 9 and 18 weeks (Appendix A). These two genera were both unaffected by any diet at 9 weeks, indicating that diet alterations, including HS diet, can also have delayed effects on the microbiome.

## 4. Discussion

In the present study, we analyzed alterations in the gut microbiome of adult mice fed HF, HS, and HFHS diets, at 9 and 18 weeks of diet (Figure 1A). A global assessment of microbiome alterations indicated a strong effect of time regardless of the diet (Figure 1B,C). It also indicated that HF induced more heterogeneous responses than HS (Figure 1D). Both microbiome richness and diversity decreased over time (Figure 1E,F). Then, we looked for statistically significant alterations in phyla and genera, especially among taxonomic groups that were stable over time under the CTL diet. We analyzed both lowly and highly abundant phyla and genera and uncovered statistically significant alterations in both extensively studied and poorly characterized bacterial genera, including uncultured ones. The percentages of several genera were decreased under the HFHS diet at 9 (Figure 2A,B, Appendix A) and 18 weeks (Figure 3A–F). Other genera were inhibited by the HF diet (with no effect of HS diet) at 9 (Figure 2C, Appendix A) and 18 weeks (Figure 4A–J). The HFHS diet could also induce an increase in the percentage of additional genera at 9 (Figure 2D–F) and 18 weeks (Figure 5A,B). In rare cases, opposite effects of the HF and HS diets were also observed at 9 or 18 weeks. At 9 weeks, the percentage of ASF356 was decreased by the HF diet and increased by the HS diet (Figure 2G). Such an effect disappeared at 18 weeks, when only the HF-related decrease was observed (Figure 4G). At 18 weeks, the *Lachnospiraceae* NK4A136 group was increased by the HF and decreased by the HS diet (Appendix A). In *Clostridium* sensu stricto 1, one diet seemed to inhibit the effect of the other. At 9 weeks, the percentage of *Clostridium* sensu stricto 1 was increased by the HF diet only, an effect not observed under the HFHS diet (Figure 2H). Such an inhibitory effect disappeared at 18 weeks (Figure 5B). At the phylum level, a diverse set of effects was observed at 9 (Appendix A–E) and 18 weeks (Appendix A–E).

Our results revealed that the time of dietary intervention had a significant influence on the microbiome profile. First, several bacterial genera responded more slowly to dietary alterations than others. *Marvinbryantia*, *Harryflintia*, *Alistipes*, *Blautia*, Eubacterium coprostanoligenes group, Eubacterium brachy group, *Lachnospiraceae* A2, *Intestinimonas*, UCG-009 of the *Butyricicoccaceae* family, and *Ruminococcaceae* Incertae sedis were significantly altered only at 18 weeks, with no significant changes at 9 weeks. Overall, more genera were affected at 18 weeks of diet than at 9 weeks. Although rodent models are widely used to study the effects of diet on the microbiome, there is little consensus on the study duration. For example, one study compared the effects of normal and high-protein diets on the gut microbiomes of rats fed for 6 weeks [28]. Another study compared the effects of glucose and fructose after a feeding period of 12 weeks [29], while in another study, mice were fed HF diets for 21 weeks before feces collection [30]. Our study mitigates this lack of consensus by testing two time points within this range.

Second, several genera were negatively affected by fat, but not by sugar. Of these, UBA1819 (Figure 2C), ASF356 (Figure 2G), *Gastranaerophilales*, and *Clostridia* UCG-014 (Appendix A) had their abundances severely decreased under the HF diet at 9 weeks, while others, including *Marvinbryantia*, *Harryflintia*, *Alistipes*, *Blautia*, Eubacterium coprostanoligenes group, Eubacterium brachy group, and *Lachnospiraceae* A2, were significantly decreased only at 18 weeks (Figure 4B–E,H–J), revealing a slower effect of the HF diet. This HF-related effect was observed as a two-stage process, in which more than 50% of the affected genera were affected only after 18 weeks (Figure 6A). Interestingly, cases in which the HF diet caused the opposite effect (an increase in genus abundance without any effect of HS diet) were not observed.

Third, the HFHS diet was able to increase the percentages of specific genera and decrease the percentages of others. HFHS increased the percentages of *Tuzzerella* and *Anaerovorax*; however, this effect was transient, being observed at 9 weeks (Figure 2E,F), but not at 18 weeks. On the other hand, the effect on *Lactobacillus* was observed at 9 (Figure 2D) and 18 weeks (Figure 5A). The HFHS-mediated increase in abundance of *Clostridium* sensu stricto 1 was slower, as it was only observed at 18 weeks (Figure 5B). Other genera, including *Akkermansia*, *Paludicola*, *Eisenbergiella*, and *Butyricicoccus*, had their percentages rapidly decreased at 9 weeks of HFHS diet (Figure 2A,B; Appendix A), while a similar effect was observed for *Intestinimonas* and UCG-009 of the *Butyricicoccaceae* family only at 18 weeks (Figure 3E,F). These findings show that, in comparison to HF, the HFHS diet induced a set of more diverse effects (Figure 6B). There was also a potential synergistic effect of the HF and HS diets in some genera. When the HF or HS diet separately led to an increased abundance of specific genera, the HFHS diet contributed to a more evident increasing effect, as observed for *Lactobacillus*, *Tuzzerella*, and *Anaerovorax* (Figure 2D–F and Figure 5A). When both HF and HS diets resulted in decreased genus abundance, the HFHS diet also resulted in an even more pronounced abundance decrease for *Akkermansia*, *Paludicola, Intestinimonas*, and UCG-009 of the *Butyricicoccaceae* family (Figure 2A,B and Figure 3E,F). These observations illustrate the diversity of dynamic effects of fat, sugar, and their conjunction in the composition of the gut microbiome.

Based on these observations, we propose that the time alterations of gut microbiome occur progressively in at least two stages. The first microbiome alterations (detected at 9 weeks) were most likely direct diet effects. The microbiome alterations detected at 18 weeks could have been driven by gut metabolome alterations caused by the microbiome alterations detected at 9 weeks (Figure 6C). This hypothesis was not tested in this study. Nevertheless, the microbiome alterations observed between 9 and 18 weeks suggest that the diet should not have been the only driving force behind the observed microbiome alterations.

The effect of different diets and dietary patterns on the composition of the gut microbiome has been extensively studied in human patients and animal models (reviewed in [31,32,33,34]). However, those data were mostly obtained from unrelated studies, in which the different diets were not simultaneously tested in identical systems and conditions. This increased the number of variables that might have contributed to the observed differences. In our study, we overcame this limitation through the simultaneous testing of the different diets.

Specific effects of high-fat and high-sugar diets on the gut microbiome have been described. Several studies have shown that sugars and sweeteners can severely affect short-chain fatty acid production, alter the integrity of the intestinal barrier, and cause inflammation [35]. In addition to decreased short-chain fatty acid production, excessive fat consumption in humans has also been related to increased production of arachidonic acid, lipopolysaccharide, and proinflammatory factors [36]. The production and consumption of gut metabolites may involve specific gut microbiome members. For example, *Bacteroides* degrades non-digestible carbohydrates, whose breakdown products are further metabolized by *Bifidobacterium* [37]. Gut microbiome alterations have been described in metabolic syndrome patients [38], type 2 diabetes mellitus associated with obesity [39], and other diseases. Many bacterial genera that were affected by the diets in our study are related to human diseases. An outstanding example is *Akkermansia*, a mucus-degrading bacterium [40] that adheres to intestinal epithelial cells and limits the permeability of the gut barrier [41]. The robustness of the gut barrier depends on the mucus layer thickness, which is modulated by the composition of the gut microbiome [42]. The levels of *Akkermansia* were generally decreased in mouse models of inflammatory bowel disease, obesity, and diabetes [43]. It was also depleted in the gut of NAFLD human patients and mouse models [44]. Its up-regulation has been proposed as a potential therapeutic tool against these diseases [43,44] and others, including cystic fibrosis and COVID-19 [45]. *Akkermansia* was also depleted from a mouse model of autism spectrum disorders [46] and had a protective effect in a mouse model of Alzheimer’s disease [47]. Additional bacterial genera affected in diseases include *Marvinbryantia*, which was increased in epileptic rats, and *Clostridium* sensu stricto 1, which was decreased in the same animal model [48]. Dysbiosis in an autism spectrum disorders rat model included decreased abundances of *Marvinbryantia* and *Butyricicoccus* [46]. The abundance of ASF356 was increased in the gut of a Parkinson’s disease mouse model, while that of *Blautia* was decreased in the same animal model [49]. It should be noted that, in our study, the abundance of ASF356 decreased under HF and HFHS diets (Figure 2G and Figure 4G), suggesting a potentially protective effect of these diets against Parkinson’s disease. As such, the overall positive or negative impact of increased or decreased abundance of specific bacterial genera needs to be thoroughly assessed. Nevertheless, these studies exemplify how our work may help to understand the connection between fat- and/or sugar-rich diets and diseases from the metabolic, neurological, and psychiatric domains.

Limitations of our study include the lack of quantification of gut microbiome metabolites. This analysis would provide not only an independent readout of diet effects, but also a connection between the bacterial genera altered at 9 or 18 weeks of diet. Within the gut microbiome, metabolites produced by specific microbes are consumed by other microbes in a network of cross-feeding interactions in which metabolites are secreted and consumed in multiple iterative steps [50]. This network can be studied both in health and in disease conditions, such as type 2 diabetes [51]. Within this network, cross-feeding interactions among specific gut microbes can be predicted and tested [52]. Several of the bacterial genera that were affected by the diets in our study produce short-chain fatty acids and other metabolites with an impact on the host. For example, the main fermentation product of *Harryflintia* is acetate [53], while *Intestinimonas* is an important butyrate producer [54]. Short-chain fatty acids could also be exploited as drugs for therapeutic modulation of the gut microbiome composition [55]. Therefore, the conjunction of microbiome and metabolome profiling of the gut will provide a more robust understanding of diet effects on host metabolism.

Here, we have shown that fat and sugar caused overall different effects with different timings on the mouse gut microbiome. Furthermore, more bacterial genera were affected at 18 weeks than at 9 weeks. Above all, our work contributes a source of bacterial entities for further studies to uncover the impact of the diet on the host.

## Figures and Tables

**Figure 1 nutrients-15-02097-f001:**
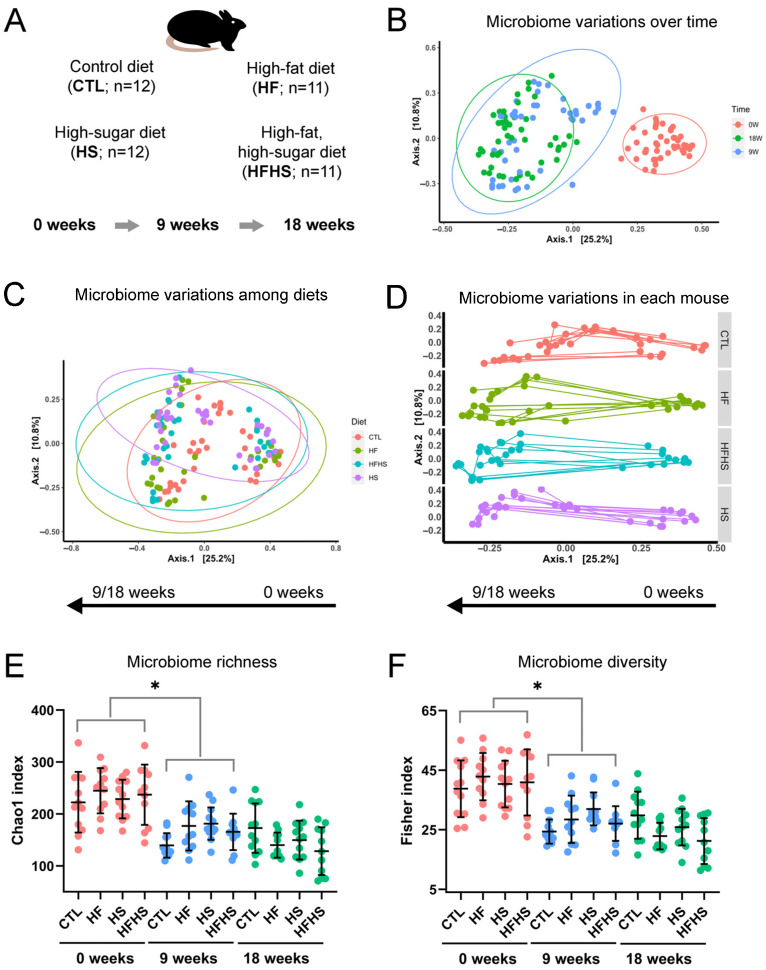
Global analysis of gut microbiome alterations. (**A**) Experimental outline. (**B**–**D**) Principal coordinates analysis (PCoA) of microbiome alterations as a function of (**B**) time, (**C**) diet, and (**D**) individual experimental mouse. In panels (**B**,**C**), ellipses represent 95% confidence intervals. (**E**,**F**) Time and diet effects on the microbiome (**E**) richness and (**F**) diversity. Error bars represent mean and standard deviation. Asterisks indicate statistically significant differences between pairs of conditions from 0 and 9 weeks (please see Appendix A for detailed statistics).

**Figure 2 nutrients-15-02097-f002:**
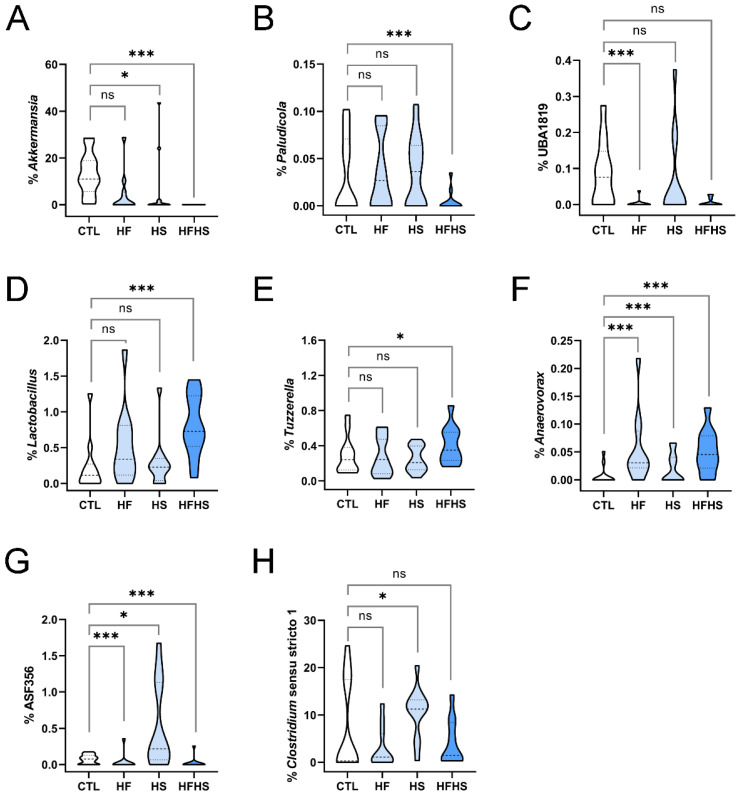
Bacterial genera whose percentages in the gut microbiome were affected by high-fat, high-sugar, or high-fat high-sugar diets, at 9 weeks of diet. Alterations are shown for (**A**) *Akkermansia*, (**B**) *Paludicola*, (**C**) UBA1819, (**D**) *Lactobacillus*, (**E**) *Tuzzerella*, (**F**) *Anaerovorax*, (**G**) ASF356, and (**H**) *Clostridium* sensu stricto 1. In each panel, the range of percentages of a specific bacterial genus under control (CTL), high-fat (HF), high-sugar (HS), and high-fat, high-sugar (HFHS) diets are represented. Dashed and dotted lines represent the median and interquartile separations, respectively. Asterisks indicate statistically significant differences and represent *p*-values adjusted using the Holm–Bonferroni method (*: 0.05 > *p* > 0.005; ***: *p* < 0.0005; ns: non-significant).

**Figure 3 nutrients-15-02097-f003:**
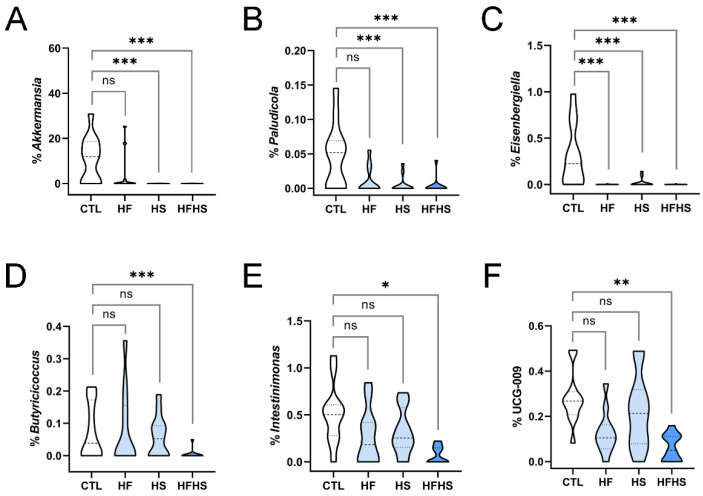
Bacterial genera whose percentages in the gut microbiome were decreased under the high-fat high-sugar diet at 18 weeks. Alterations are shown for (**A**) *Akkermansia*, (**B**) *Paludicola*, (**C**) *Eisenbergiella*, (**D**) *Butyricicoccus*, (**E**) *Intestinimonas*, and (**F**) UCG-009 of the *Butyricicoccaceae* family. In each panel, the range of percentages of a specific bacterial genus under control (CTL), high-fat (HF), high-sugar (HS), and high-fat, high-sugar (HFHS) diets are represented. Dashed and dotted lines represent the median and interquartile separations, respectively. Asterisks indicate statistically significant differences and represent *p*-values adjusted using the Holm–Bonferroni method (*: 0.05 > *p* > 0.005; **: 0.005 > *p* > 0.0005; ***: *p* < 0.0005; ns: non-significant).

**Figure 4 nutrients-15-02097-f004:**
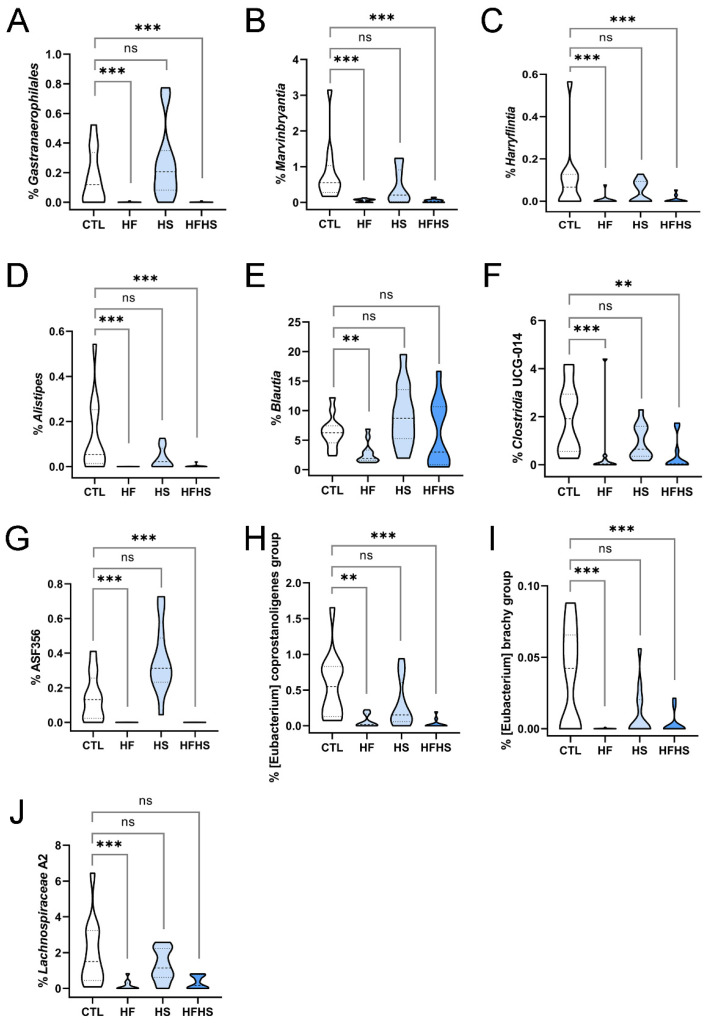
Bacterial genera whose percentages in the gut microbiome were decreased under high-fat diet at 18 weeks of diet. Alterations are shown for (**A**) *Gastranaerophilales*, (**B**) *Marvinbryantia*, (**C**) *Harryflintia*, (**D**) *Alistipes*, (**E**) *Blautia*, (**F**) *Clostridia* UCG-014, (**G**) ASF356, (**H**) Eubacterium coprostanoligenes group, (**I**) Eubacterium brachy group, and (**J**) *Lachnospiraceae* A2. In each panel, the range of percentages of a specific bacterial genus under the control (CTL), high-fat (HF), high-sugar (HS), and high-fat, high-sugar (HFHS) diets are represented. Dashed and dotted lines represent the median and interquartile separations, respectively. Asterisks indicate statistically significant differences and represent *p*-values adjusted using the Holm–Bonferroni method (**: 0.005 > *p* > 0.0005; ***: *p* < 0.0005; ns: non-significant).

**Figure 5 nutrients-15-02097-f005:**
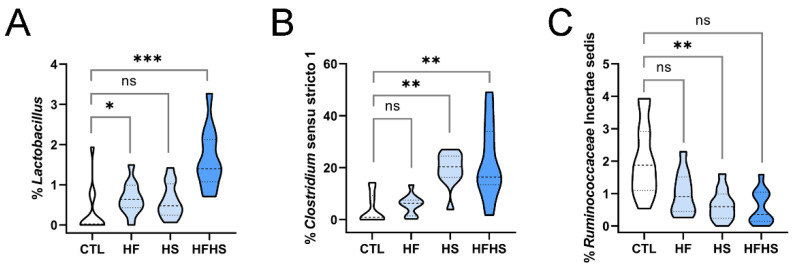
Bacterial genera whose percentages in the gut microbiome were increased under high-fat high-sugar or decreased under high-sugar diets, at 18 weeks of diet. Alterations are shown for (**A**) *Lactobacillus*, (**B**) *Clostridium* sensu stricto 1, and (**C**) *Ruminococcaceae* Incertae sedis. In each panel, the range of percentages of a specific bacterial genus under the control (CTL), high-fat (HF), high-sugar (HS), and high-fat, high-sugar (HFHS) diets are represented. Dashed and dotted lines represent the median and interquartile separations, respectively. Asterisks indicate statistically significant differences and represent *p*-values adjusted using the Holm–Bonferroni method (*: 0.05 > *p* > 0.005; **: 0.005 > *p* > 0.0005; ***: *p* < 0.0005; ns: non-significant).

**Figure 6 nutrients-15-02097-f006:**
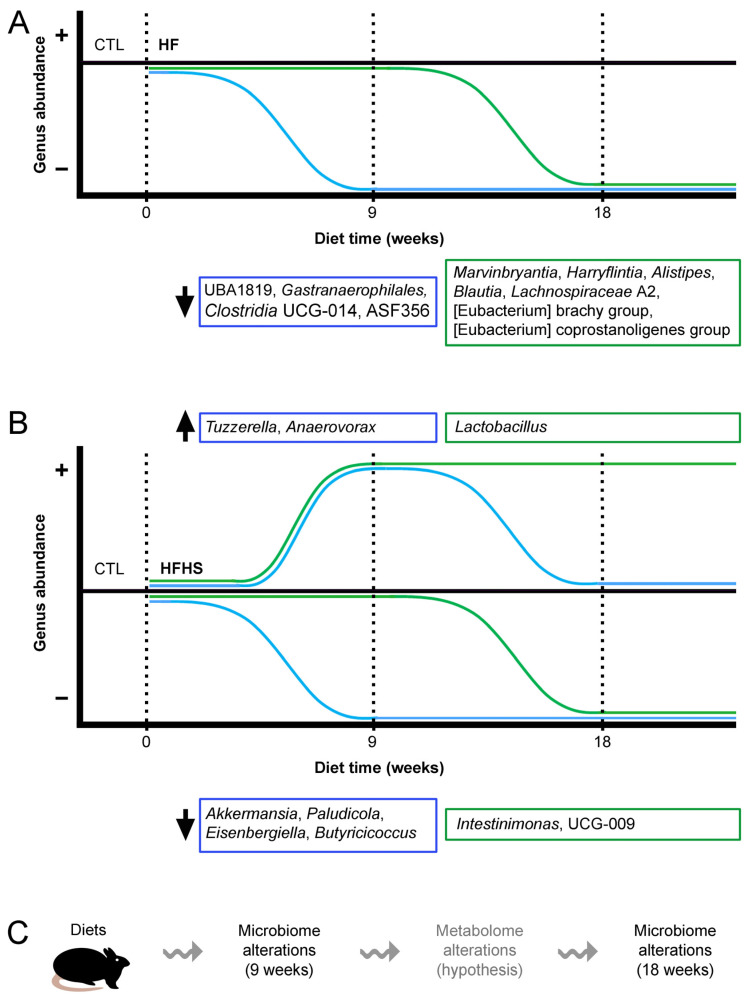
Model depicting gut microbiome alterations as a function of diet and time. (**A**) Under high-fat (HF) diet, the percentages of UBA1819, *Gastranaerophilales*, *Clostridia* UCG-014 and ASF356 were decreased at 9 weeks, an effect that was maintained at 18 weeks. At 18 weeks, the percentages of previously unaffected *Marvinbryantia*, *Harryflintia*, *Alistipes*, *Blautia*, *Lachnospiraceae* A2, Eubacterium coprostanoligenes group, and Eubacterium brachy group were also severely decreased. (**B**) Under a high-fat, high-sugar (HFHS) diet, the percentages of some bacterial genera were increased, while others were decreased. *Tuzzerella*, and *Anaerovorax* increased at 9 weeks and returned to near control values at 18 weeks. The percentage of *Lactobacillus* increased only at 18 weeks. The percentages of other genera, such as *Akkermansia*, *Paludicola*, *Eisenbergiella*, and *Butyricicoccus* decreased after 9 weeks of diet, remaining low at 18 weeks. Other genera, such as *Intestinimonas* and UCG-009 of the *Butyricicoccaceae* family, were statistically significantly decreased only at 18 weeks. (**C**) Our model predicts that, at 9 weeks of diet, the gut microbiome is altered due to direct dietary effects. Such alterations might affect the bacterial gut metabolome composition (this speculative hypothesis is written in gray), resulting in additional decreased bacterial genera at 18 weeks.

## Data Availability

Demultiplexed raw sequences are available from the Short Read Archive (SRA) under the accession number PRJNA940685.

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
