# Peer review of "The Gut Microbiome Responds Progressively to Fat and/or Sugar-Rich Diets and Is Differentially Modified by Dietary Fat and Sugar"

_nutrients, 2023, doi:10.3390/nu15092097_

Round 1

Reviewer 1 Report

Title: The gut microbiome responds progressively to fat and/or sugar-rich diets and is differentially modified by dietary fat and sugar

General comments:

The manuscript is interesting. There are published a lot of work describing the effects of high fat and high sugar diets in gut microbiota of mice models, but the current work extend current knowledge because there are several bacterial genus or species that in the past were not identified in similar trials. The science developed is good, the presentation is clear and although I am not a native English speaker, to the best of my knowledge, the English spelling and grammar is correct. I only should like to point out some minor questions of format that should be addressed in the manuscript prior to recommend its publication, as specific comments.

Specific comments:

Usually, phyla names are not written in italics and in the current manuscripts it was made. I think that it is better if the most often formats are used. Additionally, genus names should be written in italics even in the case that they be cited in the headings (lines 258-259)

Lines 59-63: This paragraph contains materials and methods and should be placed in this section. The same for the information placed in lines 144-155. This information is in most cases reiterative with those stated in Materials and Methods section. It should be unified with those stablished in materials and methods section or deleted if redundant.

Line 108: the Term “SILVA” should be defined?

Page 15, figure 6 should be reduced size.

Page 17, line 642: delete “doi”. The same in line 662 and when necessary, in other references.

References section: names of bacterial genus and species should be written in italics.

Author Response

1- Usually, phyla names are not written in italics and in the current manuscripts it was made. I think that it is better if the most often formats are used. Additionally, genus names should be written in italics even in the case that they are cited in the headings (lines 258-259)

Reply: We thank the Reviewer for this remark on text consistency. In accordance with these recommendations, phyla are now NOT written in italics in the main text (pages 6-8), in supplementary figures 1 and 3, and in supplementary tables 7 and 8 (pages 1, 3, and 10-12 of the supplementary materials file). In addition, genera were written in italics in the headings of sections 3.4-3.13 (pages 6-12 of the main text).

2- Lines 59-63: This paragraph contains materials and methods and should be placed in this section. The same for the information placed in lines 144-155. This information is in most cases reiterative with those stated in Materials and Methods section. It should be unified with those established in materials and methods section or deleted if redundant.

Reply: We thank the Reviewer for this observation to reduce unnecessary repetition in the description of our work. In accordance, we have shortened both the last paragraph of the introduction (page 2 of the main text) and the first paragraph of the results section (page 4 of the main text). Regarding the first paragraph of the results section, we have kept minimum technical information. We find that a brief repetition of the experimental outline facilitates the interpretation of the results section and is needed for referring to figure 1A, which contains a visual summary of the experimental outline for visual communication purposes.

3- Line 108: the Term “SILVA” should be defined?

Reply: We thank the Reviewer for this observation. The name of the SILVA database is not an abbreviation or an acronym. As explained in the original reference of the database (Quast et al., Nuclei Acids Res. 2013), it was inspired by the Latin word “silva”, which means “forest”. The website URL (https://www.arb-silva.de/) has been included in the materials and methods, sequence analysis, plotting, and statistics section (page 3 of the main text).

4- Page 15, figure 6 should be reduced size.

Reply: We thank the Reviewer for this remark. We have slightly reduced the size of figure 6 (page 15 of the main text).

5- Page 17, line 642: delete “doi”. The same in line 662 and when necessary, in other references.

Reply: We thank the reviewer for this observation. The doi has been deleted in references 16, 22, 23, 25, and 26. In addition, the doi has been added to reference 48, which is now reference 54 (pages 17-19 of the main text).

6- References section: names of bacterial genera and species should be written in italics.

Reply: We thank the Reviewer for this remark. Genera names have been typed in italics in references 37-38, 41-45, and 47-48, which are now references 40-41, 43-47, 53-54 (page 19 of the main text).

Reviewer 2 Report

The paper by Pessoa et al. presents an interesting research work about the effects of different diets and timings on the mouse gut microbiome.

The Introduction contains a clearly exaustive background that defines the aims of the work and its meaning. The Materials and methods section is well-described and explained. The methodologies and the data collection are adequately performed and clearly described. The figures and tables are used appropriately and the results are described comprehensively. The conclusions are supported by the results presented and highlight the importance and relevance of the study. The authors also explain all the limitations of the study.

Finally, the manuscript reports an interesting research and it appears exhaustive and to be well structured and complete.

Author Response

We thank the Reviewer for the appreciation made of our work. Since no specific alteration requests were made, our resubmitted version has addressed only the remarks from the other reviewers.

Reviewer 3 Report

The authors attempted to study how the dietary intake of sugars and fats actually alters the gut microbiome composition. To achieve this, they resorted to the mice model where they divided 46 mice into four groups with (1) the high-fat, high-sugar (HFHS) diet, (2) the high-fat (HF) diet, (3) the high-sugar (HS) diet, and (4) the control diet (CTL). They first investigated their data via the PCoA plot and alpha diversity. Then the differential analysis across four groups enables them to identify bacterial taxa that are differentially enriched. Finally, they summarized their results using a model schematic. Although the presented results are interesting, I still have some doubts regarding the presentation. I am quite open to looking at a revised version if the authors could address some major and minor issues in a satisfactory fashion, which we describe in more detail below.

Major issues:

1.     I think a more in-depth discussion of how the intake of dietary sugar and dietary fat differentially influence the gut microbes and how the metabolism of gut microbes eventually impacts metabolite production is lacking. Does dietary sugar reduce the abundance of microbes that produce short-chain fatty acids? Could authors refer to previous reports of microbe-metabolite consumption and production interactions for a clue (Tong Wang et al., PloS Computational Biology 2019; Jaeyun Sung et al., Nature Communications 2017)?

2.     I am not sure if I fully comprehend the coloring in Fig. 1D. I understand that the authors used four different colors in Fig. 1C to differentiate four types of diets provided to mice. However, in Fig. 1D, a gradual change in each color is adopted. Are they using it to show the progression of time? Or are the different degrees of colors adopted just to distinguish fecal samples from different mice? I think it is better to use the same colors for all mice at a certain time. For instance, all mouse samples at the time of 0 weeks are colored by the same color, while all mouse samples at the time of 9/18 weeks are colored by a slightly different color.

3.     I think the proposed model in Fig. 6, especially Fig. 6c, is highly speculative instead of being supported by their results. I acknowledge that their results demonstrate that the microbiome composition can be modulated by the difference in dietary intake. However, how the metabolome may be reshaped by the microbiome alternation and the metabolome can then reshape the microbiome is not demonstrated by their work. Also, their hypothesized mechanism here is the well-studied “cross-feeding interactions” in microbial ecology. I suggest they checked out the cross-feeding interactions where a microbe produces one metabolite byproduct that can be consumed by other microbes and properly summarize them here (Pedro Fernandez-Julia et al, Microbiome Research Reports 2022; Akshit Goyal et al., Nature Communications 2021). In addition, I think they need to highlight which part in Fig. 6c is supported by their evidence and which part is their speculation based on previous literature.

Minor comments:

1.     Line 150: “…mice were fed standard chow diet…” -> “…mice were fed the standard chow diet…”

2.     Line 523: “Additional bacterial genera affected in disease include…” -> “Additional bacterial genera affected in diseases include…”

Author Response

Major issues:

1- I think a more in-depth discussion of how the intake of dietary sugar and dietary fat differentially influence the gut microbes and how the metabolism of gut microbes eventually impacts metabolite production is lacking. Does dietary sugar reduce the abundance of microbes that produce short-chain fatty acids? Could authors refer to previous reports of microbe-metabolite consumption and production interactions for a clue (Tong Wang et al., PloS Computational Biology 2019; Jaeyun Sung et al., Nature Communications 2017)?

Reply: We thank the Reviewer for this observation on the discussion of our work. In the discussion section (page 14 of the main text), we mention two studies on the specific effects of fat or sugar on the gut microbiome, highlighting their impact on the levels of short-chain fatty acids (Garcia et al., Diabetology 2022; Wan et al., Gut 2019) and one of the references recommended in major issue #3 (Fernandez-Julia et al, Microbiome Research Reports 2022). Regarding the above-mentioned references, those were cited in the framework of major issue 3, in which we found that they would fit better.

2- I am not sure if I fully comprehend the coloring in Fig. 1D. I understand that the authors used four different colors in Fig. 1C to differentiate four types of diets provided to mice. However, in Fig. 1D, a gradual change in each color is adopted. Are they using it to show the progression of time? Or are the different degrees of colors adopted just to distinguish fecal samples from different mice? I think it is better to use the same colors for all mice at a certain time. For instance, all mouse samples at the time of 0 weeks are colored by the same color, while all mouse samples at the time of 9/18 weeks are colored by a slightly different color.

Reply: We thank the Reviewer for this remark on figure clarity. The slightly different colors used in figure 1D seek to highlight differences among mice from the same diet group, not the effect of time. We have edited figure 1D (page 4 of the main text) in order to have all mice within the same group identified with the same color.

3- I think the proposed model in Fig. 6, especially Fig. 6c, is highly speculative instead of being supported by their results. I acknowledge that their results demonstrate that the microbiome composition can be modulated by the difference in dietary intake. However, how the metabolome may be reshaped by the microbiome alternation and the metabolome can then reshape the microbiome is not demonstrated by their work. Also, their hypothesized mechanism here is the well-studied “cross-feeding interactions” in microbial ecology. I suggest they checked out the cross-feeding interactions where a microbe produces one metabolite byproduct that can be consumed by other microbes and properly summarize them here (Pedro Fernandez-Julia et al, Microbiome Research Reports 2022; Akshit Goyal et al., Nature Communications 2021). In addition, I think they need to highlight which part in Fig. 6c is supported by their evidence and which part is their speculation based on previous literature.

Reply: We thank the Reviewer for this remark on the rigor of our work. In the discussion section (page 15 of the main text), we briefly mention the cross-feeding interactions model and its relevance to understanding the impact of the gut microbiome on the gut metabolome. We cite one of the references recommended here (Goyal et al., Nature Communications 2021) and the two references recommended in major issue #1 (Wang et al., PloS Computational Biology 2019; Sung et al., Nature Communications 2017). We have also edited figure 6C. In order to mean that the metabolome alterations are speculative, we have written the “Metabolome alterations (hypothesis)” statement in grey, to make it less outstanding within the figure panel. In addition, in the legend of figure 6 (page 16 of the main text), we have mentioned that the (speculative) hypothesis was written in gray to distinguish it from the remaining elements of the figure, which are supported by our data.

Minor comments:

1- Line 150: “…mice were fed standard chow diet…” -> “…mice were fed the standard chow diet…”

Reply: We thank the Reviewer for this language remark. We have made the requested text correction (page 4 of the main text).

2- Line 523: “Additional bacterial genera affected in disease include…” -> “Additional bacterial genera affected in diseases include…”

Reply: We thank the Reviewer for this additional language remark. We have made the requested text correction (page 14 of the main text).

Reviewer 4 Report

The article is well structured.

Small suggestions are made:

1- Authors are requested to prepare a graphical abstract;

2- In item 2.1, why do some groups have 11 animals and others 12 animals?

3- Authors are asked to be more specific with regard to diet. It is important to show the chemical composition and more constituents present.

Author Response

1- Authors are requested to prepare a graphical abstract.

Reply: We thank the Reviewer for this recommendation, which enhances the visual communication capacity of our work. A graphical abstract has been submitted in the revised version of our manuscript.

2- In item 2.1, why do some groups have 11 animals and others 12 animals?

Reply: We thank the Reviewer for this remark. In the materials and methods, animal treatments section (page 2 of the main text), we have explained the slightly different animal numbers among the experimental groups, as follows: “Due to their disruptive behavior, one mouse had to be removed both from the HF and the HFHS diet groups, resulting in slightly different animal numbers among the four experimental conditions”.

3- Authors are asked to be more specific with regard to diet. It is important to show the chemical composition and more constituents present.

Reply: We thank the Reviewer for this observation, which is pertinent for ensuring the reproducibility of our work. We have included a more detailed description of the control and the high-fat diets in the materials and methods, animal treatments section (page 2 of the main text). On page 4 of the supplementary materials section, we have also included an additional table (supplementary table 1) containing a more detailed description of the diets and indicating their manufacturer (Mucedola s.r.l., Milan, Italy). As a consequence of the creation of this additional table, the numbers of the other supplementary tables (pages 5-12 of the supplementary materials section) were shifted by n+1 and their numbers were updated in the main text.

Round 2

Reviewer 3 Report

The authors answered all my questions. I have no further comments.

Reviewer 4 Report

The article can be accepted for publication.